# Finding Explanations in AI Fusion of Electro-Optical/Passive Radio-Frequency Data

**DOI:** 10.3390/s23031489

**Published:** 2023-01-29

**Authors:** Asad Vakil, Erik Blasch, Robert Ewing, Jia Li

**Affiliations:** 1Department of Electrical and Computer Engineering, Oakland University, Rochester, MI 48309, USA; 2Air Force Office of Scientific Research, Arlington, VA 22203, USA; 3Sensors Directorate, Air Force Research Laboratory, WPAFB, Dayton, OH 45433, USA

**Keywords:** heterogeneous sensor fusion, dense optical flow, greedy algorithm, explainable AI, canonical correlation analysis

## Abstract

In the Information Age, the widespread usage of blackbox algorithms makes it difficult to understand how data is used. The practice of sensor fusion to achieve results is widespread, as there are many tools to further improve the robustness and performance of a model. In this study, we demonstrate the utilization of a Long Short-Term Memory (LSTM-CCA) model for the fusion of Passive RF (P-RF) and Electro-Optical (EO) data in order to gain insights into how P-RF data are utilized. The P-RF data are constructed from the in-phase and quadrature component (I/Q) data processed via histograms, and are combined with enhanced EO data via dense optical flow (DOF). The preprocessed data are then used as training data with an LSTM-CCA model in order to achieve object detection and tracking. In order to determine the impact of the different data inputs, a greedy algorithm (explainX.ai) is implemented to determine the weight and impact of the canonical variates provided to the fusion model on a scenario-by-scenario basis. This research introduces an explainable LSTM-CCA framework for P-RF and EO sensor fusion, providing novel insights into the sensor fusion process that can assist in the detection and differentiation of targets and help decision-makers to determine the weights for each input.

## 1. Introduction

Even with the widescale use of neural network-based algorithms becoming common in everyday aspects of life such as advertising, banking, manufacturing, medical applications, and many more, it remains difficult to fully understand of how these algorithms utilize the data with which they are provided. Neural network-based algorithms are considered black boxes; however, unlike traditional algorithms they are considerably more versatile in the ways in which they can be trained to learn from different sources of information. In the modern age, there are an almost infinite number of potential sources of data. In the case of marketing, information from social media can be used to infer the products and services a user might be interested in based on trends that include their socioeconomic status, political views, residence, job, and many other seemingly unrelated factors.

The ability to sift through such junk data and fulfill the desired application is the main reason such black box systems are popular and widely used despite the tradeoff of users not having a complete understanding of their process. While many types of data may not be sufficient to achieve the desired objective independently, more sources of information can provide better context and trends that such algorithms can use. The fusion of multiple sources of data leads to a more robust algorithm, allowing an algorithm utilize other algorithmic tools in turn to achieve better results.

When it comes to applications such as detection, classification, and tracking, one of the tools that is especially useful is canonical correlation analysis (CCA). Canonical correlation analysis has been used in the field of sensor fusion for a number of applications in recent years [1]. A statistical method that focuses on maximizing the correlation between different datasets, it has a versatile number of applications and is a tool that other fusion models can capitalize on in order to improve overall performance. This can be especially useful when the exact relationship between different sensors is not completely known and may be more difficult to intuitively exploit.

This is particularly the case in passive radio frequency (P-RF) and electro-optical (EO) sensor fusion. Due to the nature of P-RF data, which uses radio frequency data for applications that do not directly reconstruct relative position via methods such as radar, it can be quite difficult to use. The fusion for P-RF and EO data is extremely desirable due to their complementary nature. While P-RF might not excel in the spatial resolution provided by EO sensors, it is not limited by ocular obstructions and can provide excellent range along with angular and spectral resolution. While there are a wide variety of methods to utilize EO data, even after synchronizing the two modalities it is not always apparent what method should be used to achieve the goal of fusing the two sources of information together. The use of a powerful tool used in the fields of image enhancement [2] and cognitive radio [3] naturally makes experimenting with the use of canonical correlation analysis (CCA) for EO and P-RF sensor fusion highly desirable for object detection and tracking.

However, even if a fusion model performs well for the desired task, it remains important to be able to understand its decision-making process. Neural network-based models encounter the problem of explainability, as they are essentially black box systems. It has been shown that such algorithms are not immune to bias [4], and by using explainability and visualization techniques it becomes possible to reduce the uncertainty inherent in training a black box model. Considering the possible uses for such models, including algorithmic trading, medical diagnosis, and autonomous vehicles [5], along with the major impacts they can have on their users, it is clear that models that handle such important applications should have a certain level of interpretability.

To achieve explainable sensor fusion, the present research presents CCA utilities for P-RF and EO sensor fusion for the purposes of target detection, tracking, and differentiation. We utilize a greedy algorithm and visualization methods to draw inferences about the fusion model’s decision process in order to provide greater transparency. Using the weights provided by the greedy algorithm and saliency maps from the visualizations, we infer the behaviors and mannerisms of the model to support our understandings regarding the use of fusion data. Using a modified deep CCA fusion model, we are able to confirm the value of passive RF data for this application, achieve higher performance with an explainable model, and aid in furthering the use of P-RF and EO data for target tracking. In the rest of this paper, Section 2 covers the relevant literature, Section 3 further explains our experimental design and methodology, Section 4 covers the experimental results, Section 5 provides a discussion of the results, and Section 6 concludes the paper.

## 2. Literature Review

### 2.1. Canonical Correlation Analysis

Canonical correlation analysis was first published in 1936 by Harold Hotelling, and has seen increasing use in recent years [6]. Originally proposed for association between arithmetical potentials, the algorithm quickly grew into one for finding the best predictors among linear functions by maximizing the correlation coefficient between two sets. There are a wide variety of different CCA-based approaches, including Latent Variable Model and Bayesian CCA [7] and Multiview CCA, which can be further divided into the *pairwise correlation* [8], *zero order correlation* [9], and *high order correlation* [10] approaches. In addition, there are kernel-based methods such as *Kernel CCA* [11] and *Discriminative CCA* [12], which are further divided into global and local discriminative methods, as well as Sparse CCA [13], Locality-Preserving CCA [14], and many more. However, for the purposes of this paper, the majority of the focus is on CCA applications tied to deep learning-based approaches, specifically, LSTM.

*Deep CCA* comes in many popular forms, typically using Deep Neural Networks (DNN), Autoencoders, or Convolutional Neural Networks (CNN) [15]. There are advantages to using this approach compared to approaches such as Kernel CCA or locality preserving CCA. This nonlinear approach to using CCA is not restricted by a predefined kernel or local information. Rather than relying on handcrafted features, Deep CCA and similar approaches seek more complex nonlinear associations between two views or observations by passing the information through a neural network. Considering the major impact and diversity in the methods used for multi-modal fusion, the appeal of CCA for high dimensional data becomes obvious. In previous research, our group has determined that deep learning is capable of differentiating between targets using P-RF data [16] when fused with EO data for vehicle and even human targets [17]. Thus, the use of ML with CCA is desirable to enhance performance, motivating the use of a modified deep CCA network [15] for fusion.

### 2.2. Eo/Rf Sensor Fusion

When it comes to modalities that involve radio frequency (RF) and electro-optical (EO) sensors, the focus for fusion research has traditionally been on active RF [18] sensors and [19] applications. Whether it is the fusion of Synthetic Aperture Radar (SAR) and multi-spectral images [20] via random forest classifier or double-weighted decision level neural network fusion schemes [21], many EO/RF sensor fusion applications normally use active RF in one form or another. Doppler radar and imaging radar (e.g., side-looking airborne radar) as well as other similar active RF sensors are well suited for tracking a moving target, a combination that is highly desirable when successfully fused with EO input. The combined view and the exploitation of the two types of sensor modalities has room for improvement [22], however. RF based modalities excel in providing features that are desirable for tracking. Range, angular, and spectral resolution of information from RF modalities has long been the basis of radar technology, and is adept at tracking moving targets. The benefits of combining these data with the higher spatial resolution of EO-based sensors is extremely desirable for detection, differentiation, and tracking of targets in a number of environments. There are a few RF based modalities that are used in applications such as tracking, proximity, localization, and detection. While many EO modalities are intuitively easier for humans to understand and implement for similar applications, unlike RF modalities, RF approaches to such applications are less susceptible to outside factors such as ocular interference. RF-based sensors are not limited or obscured by factors such as visual interference from natural phenomenon such as fog, clouds, snow, or any other form of weather that would otherwise normally interfere in the collection of EO data. In addition to this, RF-based sensors can provide repetitive coverage over a wide geographical area, and can determine the precise distance and velocity of a target.

There are many methods for using RF data, including both active and passive usage. In [23], support vector machine (SVM) is used as a final method of classification to achieve sense-and-avoid for unmanned aircraft. The use of an autoencoder-based dynamic deep directional unit network [24] was capable of learning compact and abstract feature representations from high-dimensional spatiotemporal data of full motion video and I/Q data for the purposes of event behavior characterization. Other research into achieving EO/RF fusion for vehicle tracking and detection using Full Motion Video and P-RF includes joint manifold learning [25], a sheaf-based approach with its data [26], and SVM classifier [23]. In [25,26], simulation data were used as the primary method of training and testing, while in [23] real data were used. In [25], a joint manifold learning fusion approach was used for mixed simulation data using DIRSIG-generated data. Finally, in [26], simulated multi-sensor data were used to locate a moving emitter with *Sheaf Theory*.

While most research is focused on active RF applications, there are a number of advantages for the implementation of *passive RF* modalities such as passive radar and RFID compared to the use of active methods. Passive RF modalities are more difficult to detect, and typically have lower power requirements and lower costs compared to the construction and use of active radar. Furthermore, they are harder to implement countermeasures against; jamming and spoofing can corrupt the collection of RF-based modalities and transmitted imagery. Combining the two modalities improves overall reliability, and has previously been implemented in a few applications for target detection, estimation, and tracking. While P-RF data have been used in a fusion-focused approach [24,27,28], if a blackbox based approach is utilized it can be difficult to understand exactly how the data are used. For this reason, having a level of transparency can be extremely helpful, which is why the main focus of the present research is on explainability with respect to how the modalities are used.

### 2.3. Explainable AI

As explainable AI (XAI) is an emerging field in research and industry, there has yet to be a widely adopted standard [29], let alone a widely adopted method of quantifying interpretability for explaining models. Even discussing the different methods of how to quantify such approaches is quite a task in itself, as there are a number of different classifications for such types of interpretability. To even begin discussing the topic of XAI, the most important thing is to define interpretability. There are several definitions of interpretability or explainability. For models that deal with EO modalities for example, interpretability might be defined as being able to map the predicted class [30] into a domain that might allow the human user to infer the decision-making process. In an ideal system, one might even define interpretability as a reasonable explanation as to why a collection of features contributed to the decision-making process, or allow for determining how much weight the decision-making process gave to said [31] features.

From saliency maps to activation maximization, there are a number of methods by which interpretability can be achieved. The distinctions between these types of methods are typically twofold, described as either ante hoc or post hoc, local or global, or model-specific or [32] model-agnostic. Ante hoc and post hoc describe an intrinsically interpretable model from different approaches. Ante hoc systems provide explanations from the beginning of the model, such as the Bayesian Rule List [33], a generative model that yields posterior distribution over decision lists consisting of a series of if–then-statements. Other examples can include methods such as visualization, saliency mask, rule extraction, and even neuron activation. Post hoc techniques, on the other hand, focus on creating explainability in a model based on the model’s outcome, marking the part of the input data responsible for the final decision. Similar to ante hoc techniques, this can include visualization and saliency mapping, and can use methods such as gradients and feature importance as well.

Similar to these, though not quite the same, are the descriptor *local* and *global* interpretability. Local interpretability provides explanations only for each single prediction, while global interpretability explains the logic of the whole system, from the input to every possible outcome. Methods such as Grad-CAM [34] are examples of local interpretability systems, using global average pooling and heat maps of a pre-softmax layer in order to determine the regions of an image responsible for prediction. Lastly, the model-specific and model-agnostic descriptors refer to the usability of different aspects of the system, with model-agnostic being indifferently usable and model-specific is tied to a particular type of black box or data. For methods that are post hoc-oriented solutions to explainability, there exist a number of methods for image-based neural networks. Visualizations, the use of gradients [35], activation maximizations [36], rebuilding the final layer of a neural network via deconvolutions [37], and applying decomposition [38] are among the common approaches. Visualization techniques typically use tools such as generative models or saliency maps in order to determine activations produced on each layer of a trained CNN or DNN after processing an image or video. The visualization of the key neurons or neuron layers highlights the responsible features that lead to a maximum activation or the highest possible probability of prediction. Deconvolution, sometimes referred to as inverting DNNs [39], can be applied to create special typical inputs or parts of an input. These special inputs are created to fit the desired output of the network, producing a special layer or single unit to recreate the results. Finally, decomposition/isolation, transfer, and limitation of portions of networks can provide further insights into which way single parts of the architecture influence the output layer.

## 3. Design and Methodology

This section highlights the components of the experimental design. The first subsection covers the dataset, the scenarios chosen, and the general objectives for detection and tracking. The second subsection covers how the chosen data types, that is, EO and P-RF, were preprocessed prior to being used for training. The third subsection covers the model utilized for the explanations and the highest-performing model that our research group tested on this dataset. The last subsection covers the greedy algorithm used for the explanations, the results of which are discussed in Section 4.

### 3.1. The Escape Dataset

In 2019, a collaboration between the Air Force Research Laboratory (AFRL) and Michigan Tech Research Institute (MTRI) released their Experiments, Scenarios, Concept of Operations, and Prototype Engineering dataset (ESCAPE) [40]. The ESCAPE dataset is a versatile toolkit of different sensor modalities and scenarios that include infrared (IR), full motion video (FMV), passive RF data, and acoustic, seismic, and active radar imagery data. This information is collected via a number of sources, with the majority of the data being collected by portable or preexisting towers that remain stationary during testing.

The primary advantages of using this dataset are the number of options for each of its scenarios and the design of the ESCAPE dataset. Multi-source data collection provides a number of vantage points from which information on various ground targets can be used for data fusion research. The design of the dataset enhances the complexity and opportunity of such research by increasing the number of available modalities, in addition to outdoor experimental irregularities. In this dataset, various ground vehicles are witnessed leaving the observed scene available sensors, then potentially reemerging, thereby *“escaping”* detection and tracking. This design incentivizes ML algorithms to utilize other sensory data in order to confirm target detection, tracking, and classification. A brief description of the scenarios we used can be found in Table 1.

There are a total of five different types of ground vehicles used in the dataset: a gas motor Gator utility vehicle, a diesel motor Gator utility vehicle, a pickup truck, a panel van, and a stake rack truck. It should be noted that between the Gator vehicles the diesel-powered gator had different acoustical and seismic signatures due to the nature of its propulsion system, despite how relatively similar the John Deer vehicles look compared to the trucks or vans. These five vehicles are the primary focus of the ESCAPE dataset, and by design are always the aforementioned targets of the dataset.

For the multimodal heterogeneous EO/P-RF sensor fusion research presented in this paper, the raw RF data were preprocessed to obtain I/Q histograms with respect to the time. The histograms were then aligned with the simulated EO data for the purposes of detecting and discriminating between the different vehicles in each scenario. These P-RF data were obtained from three sources, designated as points 11, 12, and 13 for the MTRI P-RF sensors (depicted in Figure 1 as the orange trapezoids 11, 12, and 13) used to collect the I/Q data. The EO data were collected by MTRI EO Sensor 04, shown in Figure 1 as the blue semicircle. While the ESCAPE dataset has a total of nine scenarios, for the purposes of experimentation the research presented in this paper uses Scenarios 1, 2(2C), and 3(2D). These scenarios were picked specifically for a number of reasons. Based on results from earlier research with the same dataset, the EO input provided by the two SUAS produced less than acceptable results in terms of accuracy, while the MTRI EO sensor designated, designated as Sensor 04, provided the optimal results.

### 3.2. Data Preprocessing

In order to better exploit the combined view that the P-RF and EO sensors provide, certain steps had to be taken in order to implement their fusion. From previous research [7] with this dataset, we determined that the input of the raw P-RF data by itself was not sufficient for fusion, even via neural network. There had been other attempts using supervised learning and other classifiers. However, the performance of these classifiers was insufficient even for classifying scenarios, let alone handling the task of identifying specific targets. While the ESCAPE dataset does provide the sensory information required for radar, this research is primarily focused on exploitation of the *passive* RF information available in concert with the EO data. Because the raw in-phase and quadrature components (I/Q data) were insufficient for classification purposes, the data were transformed into a series of histograms corresponding to the same points in time as each of their respective frames. The EO data, on the other hand, were preprocessed through dense optical flow (DOF) to remove inactive targets from detection. In prior research [16], it was determined that the use of a neural network-based approach is desirable, as when traditional methods were compared all except *nearest centroid* performed underwhelmingly. To handle this crucial issue of computer vision, dense optical flow (DOF) [41] was applied on the EO images. Lastly, the application of canonical correlation analysis and the input of the variates between the P-RF and EO data was a necessary step for the creation of the current fusion model. In previous research, the application of CCA variates drastically improved the performance of the classifiers for discriminating between different targets. In the case of P-RF and EO fusion for the ESCAPE dataset, the results of the XAI indicate that the variates provide insight that can be almost as valuable as the DOF-EO input. As previous research shows that including the canonical variates is proven to increase performance with respect to F1 score and the Tracking Detection Rate, the use of CCA variates in training for the fusion model was a clear choice for the XAI research.

### 3.3. Lstm-CCA

For the purposes of fusion, the model utilized was an *LSTM-CCA* as seen in Figure 2 below. This model uses the same inputs as the data frame fusion model for explainable AI, and can therefore provide a fair comparison in order to determine whether the greedy algorithm prototype can perform competitively with a less *transparent* ML model. The LSTM-CCA model uses a CCA layer derived from the work of *Deep CCA* [42]. While LSTM-CCA is often used for prediction of time-series data such as medical applications [43] or fault monitoring [44], the nature of the “evasion” of the targets makes it extremely useful for this dataset. The Deep CCA creates a layer that computes the representations of the two views (P-RF histograms and DOF-EO frames for corresponding points in time) that are connected from two deep networks. While the neural network type is different, using the same architecture to process the two views separately with CCA is a desirable approach for this dataset based on the target detection and classification results. Thus, the outer layers were trained to be maximally correlated with each view.

LSTM-CCA relies on the CCA Layer approach of Deep CCA with a few minor changes. The correlations of the two outermost views are used, then the correlations and two views are fed into an LSTM. These results are then run through the network in sequence and used to produce a classification output to determine which vehicles are moving during the provided frames. After the training is completed, evaluation begins. For the purposes of implementation, the equations remain the same, with a change in optimization function to RMSProp followed by a standard sigmoid activation function. Rather than implementing the system only with deep neural networks, the resulting views are saved with respect to time, being loaded as a sequential vector for the LSTM portion of the network. While the neural network architecture and implementation of the original Deep CCA layer differ, the purpose of the CCA layer’s implementation in the LSTM remains the same. The model then undergoes training and cross-validation, with thirty percent of the data retained for validation and seventy percent used for training.

### 3.4. Explainable AI

For the purposes of experimentation, the analysis of the LSTM-CCA Fusion model occurs after preprocessing and compares the local and global weights of the fusion model for each of the scenarios with respect to their performance with individual targets. To facilitate this, explainX.ai is used. ExplainX.ai uses a streamlined and optimized version of [45] ProtoDash, a versatile algorithm that works with any black box ML algorithm to identify similar prototypes. The creation of these prototypes, representatives that optimally describe the black box algorithm, allow for the coherent framework to find and determine non-negative weights by importance. Using this framework combined with the current data frame and CCA inputs is what composes the generated Greedy Algorithm used in the experiments.

ProtoDash uses a greedy algorithm to achieve optimization, seeking to assess the importance of the generated prototype and using the nonnegative weights in order to produce a more natural and easier to interpret comparison. The prototype framework focuses primarily on deriving the theoretical bounds for the selection methods. The 2019 algorithm showcases its actionability, utility, and insight when summarizing a variety of different datasets and applications (MNINST, Retail, CDC questionnaires case study). ExplainX.ai is based off of this algorithm, though the number of available features it provides is considerably increased, and is capable of providing insights based on a single prediction point as well as providing a global overview of the interactions of different features in a *user-friendly* manner. With respect to the features that explainX.ai provides, the four major categories of transparency are global explanation, feature interaction, distribution, and local interaction. For the purposes of this paper, the primary focus is on global explanation and the comparison of weights in decision making. Though certain aspects of local interaction are brought up in Section 4, the primary results and interpretation are focused on global explanation due to the limited number of features used in the current data frame fusion model. In future work, other aspects might be integrated after the data frame fusion model is expanded, as discussed in Section 5. Global feature impact identifies which features in a dataset have the greatest positive or negative effect on the outcomes of an ML model. The impact value is the weight by which the input provided is used to produce a decision. For the purposes of this paper, the impact of different variables on the decision-making of the XAI fusion model is the focus. Due to the fact that the calculations of the feature importance and feature impact remain the same for the four sources of input data, the discussion of weight on decision-making is focused only on feature importance. In future work, depending on the results of more than four types of information placed into the model, other aspects of the global explanations might be implemented.

*Feature interaction* contains a number of visual representations of the model, specifically a partial dependence plot and a summary plot. It decomposes the predictions into different terms: a constant term, a term for the first feature, a term for the second feature, a term for the interaction between two features, etc. The interaction between the two features is the change in the prediction that occurs by varying the features after considering individual feature effects. The partial dependence plot shows the marginal effect one or two features have on the outcome of an ML model, while the summary plot provides the first indications of the relationship between a value of a feature and the impact on the prediction, with different colors representing the value of the features from low to high.

*Distribution* provides the option of viewing the impact of different variables via histogram or violin plot and the option to implement a multi-level Exploratory Data Analysis (EDA) based on the chosen variables. Histograms of the different features can be individually produced. For the joint violet plots, the statistics summary provides mean, median, model, interquartile values, etc. The distribution of the predicting variable can be found on top of the other input variables in order to find the join distribution.

Finally, *local interaction* provides the options to view local feature impact and similar profiles to the data. Local feature impact narrows down the global feature impact graph, showing the decision plot and how much each feature contributes to the overall model prediction for a specific point. Profiles generate similar profiles from within the dataset based on how similar their attributes are with respect to model prediction and ground truth values.

## 4. Experimental Results

### 4.1. Explainable AI

With respect to the performance of the tracking detection rate metric and the F1 score, the results for the XAI fusion model were able to meet our expectations and achieve an F1 score of 1.0 in both metrics for each respective target. While not practical in a larger-scale application, for the purposes of the ESCAPE dataset and the chosen scenarios this improvement makes it the ideal model to analyze with explainability and gain insights as to how the P-RF data is used. When compared with other approaches from previous research [27,28], the use of the CCA layer provided the best performance possible for the scenario data. While there is a slight change in distribution between all of them based on the target being tracked, the results from the XAI indicate that the CCA input greatly improves the heterogenous EO/P-RF sensor fusion.

As the results from the LSTM-CCA and the current XAI data were both able to achieve a perfect F1 score and perfect score for the tracking detection rate metric, the primary focus for this research is on the results for the XAI. More specifically, we focus on the weights and transparency of the current fusion model and attempt to gain insights on how the less intuitive data, that is, the P-RF data, are utilized. The use of the greedy algorithm in order to derive a prototype approximation for the fusion model provides benefits for determining the impact of different inputs for different scenarios. For the experimentation, the four inputs for the data frame were the DOF-EO frames, the EO-CCA input, the P-RF histogram input, and the RF-CCA input. For Scenarios 1, 2(2C), and 3(2D), Table 2, Table 3 and Table 4 below provide the explainX.ai results with respect to the weights of each of the four inputs.

The overall results of the weights in the experiment with explainX.ai demonstrate a reasonably well-spread and balanced result from the prototype. From the weights, it is clear that the P-RF data play something of a role within the fusion model, even if the histogram input is generally the lowest input weight relative to the other three data inputs. It is less than surprising that the EO input consistently receives the highest weight, though it is an interesting scenario in which the P-RF CCA variates are occasionally almost as important to the decision-making process as the EO. The fused view performs better when the focus is on Vehicle 1, which in the context of screen time in Scenario 1 makes sense with respect to the weight distribution.

For the most part, the results for scenario 2(2C) were within expectations. The EO input predominantly leads in terms of weight for decision-making, followed closely by the P-RF CCA values. That said, there are brief unexpected instances of the EO-CCA covariate input dominating the results of the weights over the P-RF CCA covariate input. In light of the importance of the EO input with regard to detecting the targets, it should not be a surprise that the covariate input of that data should be important as well. However, as seen in Table 2, the EO-CCA covariate input being used more in the decision-making for certain vehicles is clearly not a fluke, and in certain cases is a major discriminating feature for decision-making.

In the case of Table 4, Vehicle 1 and Vehicle 3 both rely more on the EO-CCA input. In comparison, Vehicle 5 barely uses the P-RF histogram or EO-CCA input at all. Compared with the other targets, Vehicle 5’s classification relies almost entirely on the P-RF CCA and DOF-EO inputs. Considering Vehicle 5 spends the least amount of time visible with respect to the EO source, this indicates that the P-RF CCA input provides enough insight into the activities of Vehicle 5 to be given the same weight as the EO input. While the P-RF CCA data impact is relatively high for the scenario compared to the P-RF data impact, the sheer difference in impact weight is especially visible with Vehicle 5’s weights.

### 4.2. Inferences from Explainx AI

As both the XAI and the LSTM-CCA models were both capable of achieving a perfect F1 score and tracking score of 1, the value in this experiment comes from the ability to see how the XAI used information fed from the data frame for decision-making. For each of the scenarios and each of their respective targets, there are corresponding values for the weights assigned by each of the four inputs, namely, the DOF-EO, EO-CCA variates, P-RF histograms, and the RF-CCA variates, for decision-making. The tables, however, do not address the bigger picture, instead focusing on the impact each of the individual aspects of the data frame had on the classification of the vehicles.

In order to gain further insight on the model, we first explored the local feature impact, as seen in Figure 3. The P-RF histograms appear to have a negative impact on the outcome on the prediction process. In comparison however, the RF CCA variates appear to have the highest positive impact, surpassing the DOF-EO image input and the EO CCA variate inputs. These results are consistent with the experiments in this paper, as the P-RF histograms never consistently reached performance near that of DOF-EO. The CCA input drastically improved the results of the LSTM-CCA and the generated Greedy Algorithm, which makes sense based on the the RF CCA variates bridging the gap between the P-RF histograms and accurately detecting and tracking the individual targets.

Next, we determined whether there is an overall impact of the RF CCA variates on the local feature impact graph claims. By averaging the weight values for each vehicle and each scenario, Figure 4 shows that the RF CCA input for the ESCAPE dataset holds the second-highest value after the DOF-EO input. The average weight value is slightly higher than that of the EO CCA variates, and naturally the P-RF histogram inputs are the lowest on average. The average weight of the P-RF data is as expected, with the P-RF weight almost always the lowest, with the sole exceptions of Vehicle 2 in Scenario 1 and Vehicle 4 in Scenario 3(2D), in which the RF-CCA input and the EO-CCA values are lower in weight.

Taking the results on a scenario-by-scenario basis and averaging the values for each of them, Figure 5 displays the results for Scenario 1. As seen above, the difference in average weight between the DOF-EO input and the RF CCA input is considerably close. The RF CCA data have a little more than a 0.025 difference in weight for decision-making compared to the DOF-EO input. It should be noted, however, that in Scenario 1 the EO CCA input average weight has little more than a 0.006 difference from that of the RF CCA input compared to the average, suggesting a less than 0.02 difference between the two inputs. For Scenario 1, the CCA input is almost on par with that of the DOF-EO input on average.

As for scenario 2(2C), the difference is not as close to that of the averaged results as in Scenario 1. As seen in Figure 6, there is an over 0.4 difference in average weight input from the DOF-EO input to the RF CCA variate input. The EO CCA variate input is not far behind the RF CCA’s average weight, with the results even closer than in Scenario 1. As is a reoccurring theme in this experiment, the P-RF histogram input remains at the lowest average weight. The focus on inputs for decision making in the generated Greedy Algorithm remains on the DOF-EO input followed by the RF and EO CCA variate inputs.

Finally, with respect to Scenario 3(2D), the most chaotic of the three scenarios, there were a few notable anomalies in the standalone experiments. As seen in Figure 7, the most pleasant of which was the clustering performance, as KNN just manages to exceed 0.9 accuracy for Vehicle 1 in scenario 3(2D). For this scenario and in future work revisiting the performance for KNN warrants the use of activation maximization or another form of visualization to determine the impact of the P-RF data on Vehicle 1. The overall performance of the P-RF data is much stronger in Scenario 3(2D) than in Scenarios 1 and 2(2C). While there are no other standalone instances of P-RF scoring above a 0.9, there are many that are above 0.8 in comparison to the other two scenarios.

## 5. Discussion

### 5.1. Fusion Comparison

As the simplest way for the model to determine if a potential target is present is clearly the usage of the EO data, the available data needed to be designed to “incentivize” the model’s usage of the P-RF data. This is reflected in the weights for the most part, as the EO and EO-CCA input is predominantly focused on weight wise, while P-RF and RF-CCA is less often used. However, the level of accuracy the model possesses is an important factor, as more errors might lead to a misunderstanding in how and when the P-RF data is best utilized. For that reason, using the highest performing model is integral. With respect to the performance of the tracking detection rate metric and the F1 score, the results for the LSTM-CCA fusion model were able to meet the expectations and achieve a perfect score in both metrics for each respective target. These results indicate that the CCA values and the data frame fusion model together are able to sufficiently track and detect the vehicles with the EO and P-RF data for the application and dataset in question. While there was a slight change in distribution between all of them based on the target being tracked, the results from the generated Greedy Algorithm indicate that the CCA input helps to improve the heterogenous EO/P-RF sensor fusion.

### 5.2. Explainable AI

As the results from the LSTM-CCA and the current XAI data were both able to achieve a perfect F1 score and perfect score for the tracking detection rate metric, the primary focus for this research is on the *explanations* provided by the generated Greedy Algorithm, more specifically, the weights and transparency of the current fusion model. The use of the greedy algorithm to derive a prototype approximation for the fusion model provides benefits for determining the impact of different data inputs for different scenarios, which provides context for when and where data was more useful with respect to the desired target. For the experimentation, the four inputs for the data frame were the DOF-EO frames, the EO-CCA input, the P-RF histogram input, and the RF-CCA input. For Scenarios 1, 2(2C), and 3(2D), Table 2, Table 3 and Table 4 provide the explainX.ai results with respect to the weights of each of the four inputs.

The overall results of the weights experiment with explainX.ai demonstrate a reasonably good spread and a balanced result from the prototype. From the weights, it is clear that the P-RF data play a role within the fusion model, even if the histogram input is generally the lowest input weight relative to the other three data inputs. It is less than surprising that the EO input consistently receives the highest weight, though an interesting scenario in which the P-RF CCA variates occasionally may be almost as important to the decision-making process as the EO. The fused view performs better considering the focus on Vehicle 1, which in the context of screen time in Scenario 1 makes sense with respect to weight distribution.

For most part, the results of Scenario 2(2C) were within expectations. The EO input predominantly leads in terms of weight for decision-making, followed closely by the P-RF CCA values. That said, there are brief unexpected instances of EO-CCA covariate input dominating the results of the weights over the P-RF CCA covariate input. Considering the importance of the EO input with regard to detecting the targets, it should not be a surprise that the covariate input of these data is important as well. As seen in Table 3, the EO-CCA covariate input being used more in the decision-making for certain vehicles is clearly not a fluke, and in certain cases is a major discriminating feature. In the case of Table 4, Vehicle 1 and Vehicle 3 both rely more on the EO-CCA input; interestingly enough, Vehicle 5 barely uses the RF histogram or EO-CCA input at all. Between the five targets, Vehicle 5’s classification relies almost entirely on the P-RF CCA and DOF-EO inputs. Considering that Vehicle 5 spends the least amount of time on the EO source, this indicates that the P-RF CCA input provides enough insight into the activities of Vehicle 5 for it to receive the same weight as the DOF-EO input.

### 5.3. Comparison of Weights

For each of the scenarios and each of their respective targets, there are corresponding values for the weights each of the four inputs, namely, the DOF-EO, EO-CCA variates, P-RF histograms, and the RF-CCA variates with respect to decision-making. The tables, however, do not address the bigger picture, instead focusing on the impact each of the individual aspects of the data frame has on the classification of the vehicles. To this end, in order to gain further insight on the generated Greedy Algorithm, the first thing to do was to explore the local feature impact for each of the three scenarios, as seen above. For Scenario 1, the impacts for Vehicle 1 were higher than for Vehicle 2, with Vehicle 1 prioritizing the P-RF CCA and EO inputs over the P-RF and EO-CCA inputs. Vehicle 2, on the other hand, prioritized the input of the EO-CCA data and the P-RF data, which is to be expected given the relative lack of “screen time” that the vehicle in question had on the training data. The only notable outlier with the Vehicle 2 data is the considerably lower importance weight at the local level for the P-RF CCA input compared to Vehicle 1.

For Scenario 2(2C), the local feature importance of the P-RF and EO-CCA becomes clear for almost all three of the vehicles. Vehicle 1 is the only instance for this scenario in which the P-RF CCA and EO supersede the other two inputs, with P-RF and EO-CCA being relatively tied. For Vehicle 2, the roles are reversed, with the P-RF and EO-CCA having the higher respective weights and P-RF CCA and EO having the lower local weights instead. This repeats for Vehicle 3, which makes sense due to the view for Vehicle 1 being more dominant with respect to the “screen time” compared to Vehicles 1 and 2.

Finally, for scenario 3(2D), the scenario with the largest number of vehicles, there is an outlier or two that occurs with these results for the local level feature weights. Vehicle 5 noticeably has a negative impact with respect to the EO input, which for the local data and the global data is an anomaly. Vehicle 1 primarily focuses on the EO-CCA results, with P-RF having the second highest weight, while the P-RF information has the lowest local value. This is not the case for Vehicle 2, as the P-RF CCA and P-RF data are relied on more locally, similar to how Vehicles 3 and 4 focus on the P-RF CCA and then the EO input over the other features by comparison. For Vehicles 3 and 4, the priority of P-RF CCA, EO, then EO-CCA and P-RF repeat, while Vehicle 5 prioritizes the P-RF CCA and P-RF data. It is not all that surprisingly that Vehicle 5 focuses more on changes in the P-RF data and actually assigns a local negative value for the EO, as from the EO point of view the vehicle is nonexistent, being a negative influence for the data. In comparison to Vehicles 3 and 4, which are similar, the same priorities at the local level are more focused on the EO data.

The next thing to consider is whether the RF CCA variates have as much of an overall impact as the local feature impact graph claims. By averaging the weight values for each vehicle and each scenario, the ESCAPE dataset the RF CCA input holds the second highest value, after the DOF-EO input. The average weight value is slightly higher than that of the EO CCA variates, and naturally the P-RF histogram inputs are the lowest on average. The average weight of the P-RF data is easily expected, as the P-RF weight is almost always the lowest, with the sole exceptions of Vehicle 2 in Scenario 1 and Vehicle 4 in Scenario 2(2D), in which the RF-CCA input and the EO-CCA values are lower in weight.

Taking the results on a scenario-by-scenario basis and averaging out the values for Scenario 1, the difference in average weight between the DOF-EO input and the RF CCA input is considerably closer. The RF CCA data has a little more than a 0.025 difference in weight for decision-making compared to the DOF-EO input. It should be noted, however, that in Scenario 1 the EO CCA input’s average weight has little more than a 0.006 difference than that of the RF CCA input compared to the average, which would suggest a less than 0.02 difference between the two inputs. For Scenario 1, the CCA input is almost on par with that of the DOF-EO input on average.

As for Scenario 2C, the difference is not close to that of the averaged results in Scenario 1. There is an over 0.4 difference in average weight input from the DOF-EO input to the RF CCA variate input. The EO CCA variate input is not far behind the RF CCA’s average weight, and is even closer than in Scenario 1. As is a reoccurring theme in this experiment, the P-RF histogram input remains at the lowest average weight. The focus on inputs for decision making in the generated Greedy Algorithm remains on the DOF-EO input followed by the RF and EO CCA variate inputs.

Lastly, for Scenario 2(2D), which holds the highest number of potential targets between the three, the results show that in terms of average weight distribution the DOF-EO input remains the largest impact on the decision-making process. RF CCA variates have the second highest average weight, followed by a larger gulf between RF CCA variates and the EO CCA variates than in Scenario 2C. The results in indicate that, as always, the P-RF histogram input average weight remains the lowest of the four main features provided by the data frame input. From these results, it appears that while the EO CCA variates input does have moments in which its weight is higher than that of the RF CCA variates, globally and on average the decision-making is focused on the DOF-EO image input, then the P-RF CCA variates, followed by the EO CCA variate inputs, and finally the P-RF histograms.

## 6. Conclusions

Through the use of explainX.ai (greedy algorithm), we received insights on the LSTM-CCA model with respect to the impact of the data inputs used. P-RF data collected from the I/Q data passively collected without any signals are generally not used, especially for target identification and detection, unless the situation incentivizes the use of the data for confirmation or detection. This research provides an understanding of how P-RF data can contribute to vehicle tracking and detection within an urban environment and showcases the results using real data from the ESCAPE dataset. More specifically, it provides insights into the black box necessary to utilize this information via greedy algorithm. From these results, we can infer and demonstrate the impact of P-RF and CCA data in the decision-making process of the LSTM-CCA model.

While the model predominantly utilizes EO data for its decisions, in cases where the EO data are insufficient, either due to there being a duplicate vehicle or the vehicle being out of sight of the EO source, the model relies more on the P-RF data. These results support our understanding of how the P-RF and EO data are utilized in different scenarios and provide more transparency for the importance of their respective impacts in decision making. While the target tracking application of a smaller dataset is relatively simpler to work with, having only a maximum of five targets at a time, the larger question is how to best measure the impact of the passive RF data with respect to detection and tracking. The results in terms of the P-RF histograms average weight in prediction and decision making, combined with the visualizations, indicate a more intuitive decision-making process that the fusion model follows for target differentiation and tracking.

## Figures and Tables

**Figure 1 sensors-23-01489-f001:**
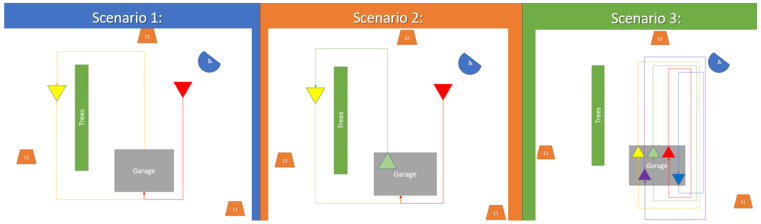
Scenarios 1, 2, and 3. In Scenario 1, Vehicle 2 (the yellow triangle) attempts to hide behind the treeline, while Vehicle 1 (the red traingle) drives around the garage. In Scenario 2, a third vehicle (the green triangle) attempts to “substitute” itself for Vehicle 2 after Vehicle 2 enters the garage. In Scenario 3, there is a shuffle as multiple vehicles, including 4 and 5 (the purple and blue triangles, respectively) begin to shuffle as they drive around and enter the garage.

**Figure 2 sensors-23-01489-f002:**
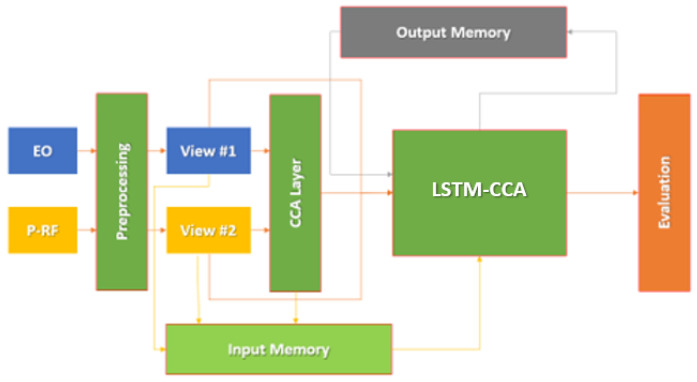
LSTM-CCA architecture overview.

**Figure 3 sensors-23-01489-f003:**
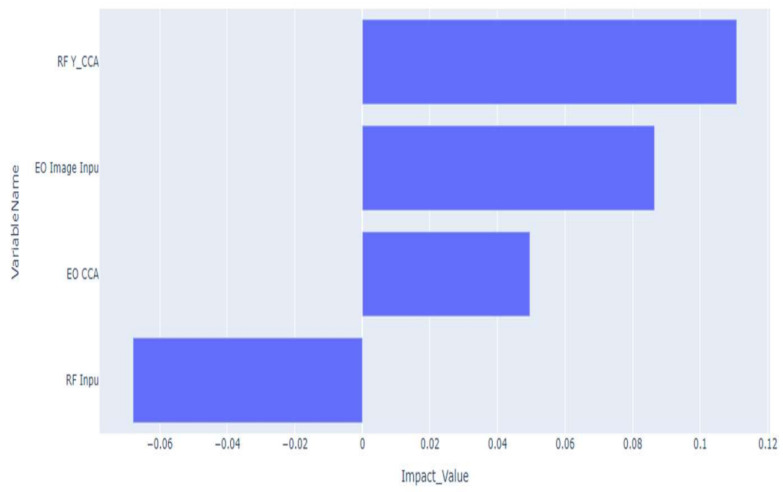
Local impact of data frame attributes for EO, P-RF, and CCA variates for EO and P-RF input data.

**Figure 4 sensors-23-01489-f004:**
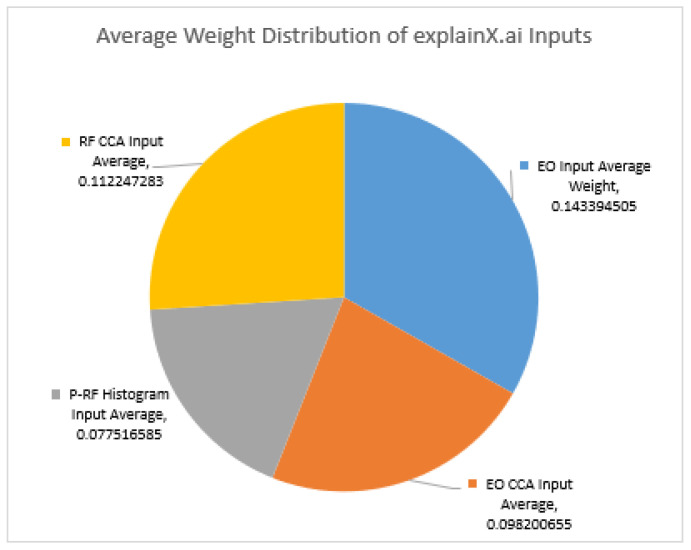
Comparison of average weights for all five targets.

**Figure 5 sensors-23-01489-f005:**
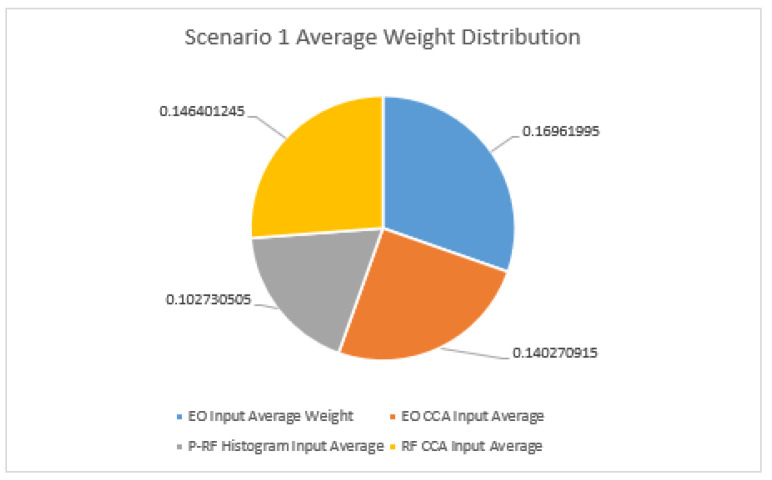
Comparison of average weights for Scenario 1.

**Figure 6 sensors-23-01489-f006:**
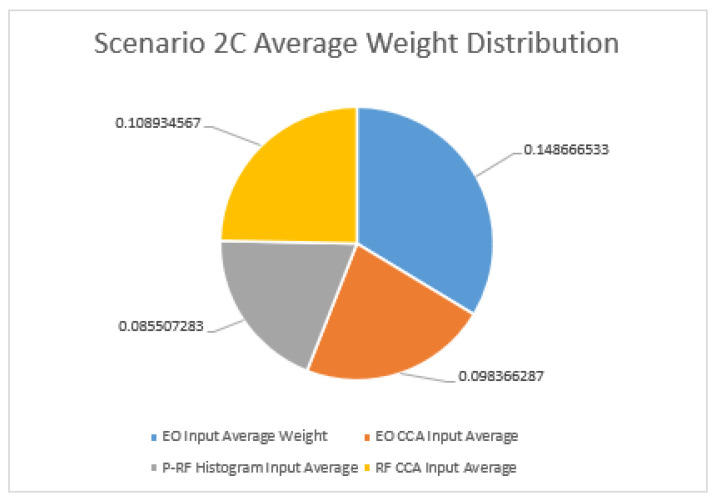
Comparison of average weights for Scenario 2C.

**Figure 7 sensors-23-01489-f007:**
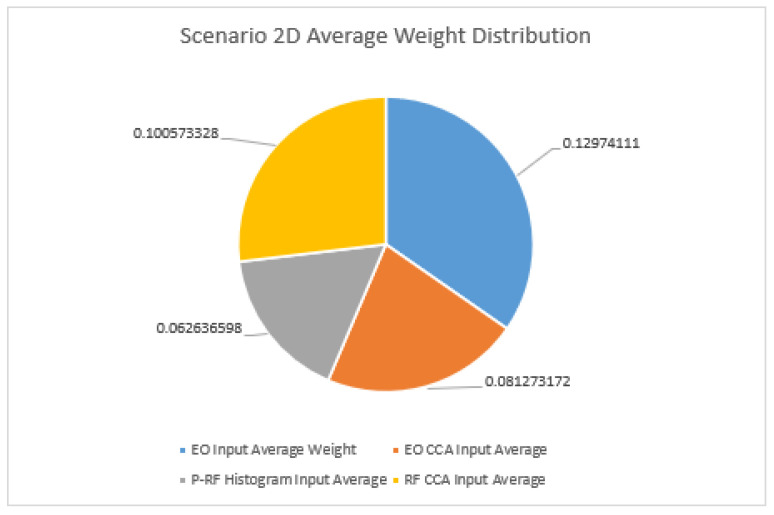
Comparison of average weights for Scenario 3(2D).

**Table 1 sensors-23-01489-t001:** Scenario Overview.

Scenario	Description
Scenario 1	Two vehicles: one behind treeline, one in sight, “switches” in the garage
Scenario 2(2C)	Three vehicles: one behind treeline, one vehicle that looks different that simply moves out of sight, one in sight, “switches” in the garage
Scenario 3(2D)	Five vehicles: four come out of the garage and shuffle the order while another comes out from the back of the garage

**Table 2 sensors-23-01489-t002:** explainX.ai weight results for Scenario 1.

Vehicle	Input	Weight
Vehicle 1	EO Input	0.212249
	EO CCA Input	0.1844515
	RF Input	0.1246046
	RF CCA Input	0.2122194
Vehicle 2	EO Input	0.1269909
	EO CCA Input	0.09609033
	RF Input	0.08085641
	RF CCA Input	0.08058309

**Table 3 sensors-23-01489-t003:** explainX.ai weight results for Scenario 2C.

Vehicle	Input	Weight
Vehicle 1	EO Input	0.1313921
	EO CCA Input	0.1134093
	RF Input	0.09182034
	RF CCA Input	0.100984
Vehicle 2	EO Input	0.1841983
	EO CCA Input	0.07001276
	RF Input	0.06620991
	RF CCA Input	0.1185514
Vehicle 3	EO Input	0.1304092
	EO CCA Input	0.1116768
	RF Input	0.0984916
	RF CCA Input	0.1072683

**Table 4 sensors-23-01489-t004:** explainX.ai weight results for Scenario 2D.

Vehicle	Input	Weight
Vehicle 1	EO Input	0.09108965
	EO CCA Input	0.0.07937257
	RF Input	0.05895651
	RF CCA Input	0.0654112
Vehicle 2	EO Input	0.10578873
	EO CCA Input	0.09031994
	RF Input	0.06189732
	RF CCA Input	0.1057738
Vehicle 3	EO Input	0.1332544
	EO CCA Input	0.1116768
	RF Input	0.0984916
	RF CCA Input	0.09032434
Vehicle 4	EO Input	0.1841983
	EO CCA Input	0.0825
	RF Input	0.08589286
	RF CCA Input	0.09032434
Vehicle 5	EO Input	0.1343745
	EO CCA Input	0.04249655
	RF Input	0.007944699
	RF CCA Input	0.134089

## Data Availability

Not applicable.

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
