# Peer review of "Finding Explanations in AI Fusion of Electro-Optical/Passive Radio-Frequency Data"

_sensors, 2023, doi:10.3390/s23031489_

Round 1

Reviewer 1 Report

I think that this paper is an interesting paper that fuses information from P-RF and EO and handles it in CCA.

The general content of the paper will be judged to be novel and effective.

But some minor corrections are required.

1. Reference [1] is not cited in the paper, so please correct it.

2. Inadequate caption for Figure 1. Also, the resolution is poor and the characters are hard to see.

3. The resolution of Figure 3 is also poor and difficult to see, so please correct it.

4. Line372-377, the vehicle 5 appearing in the description of Table 3 is confusing, so please correct the description.

5. There is no explanation for Table 4 and Figure 4 in the text. Please add an explanation where appropriate.

Author Response

Reviewer #1:

  1. Reference [1] is not cited in the paper, so please correct it.

Understood, thank you.

  1. Inadequate caption for Figure 1. Also, the resolution is poor and the characters are hard to see.

That has been fixed, please let me know if the new caption and image are sufficient.

  1. The resolution of Figure 3 is also poor and difficult to see, so please correct it

That has been fixed, please let me know if the new caption and image are sufficient.

  1. Line372-377, the vehicle 5 appearing in the description of Table 3 is confusing, so please correct the description.

I made a mistake in table citation, which is related to 5), and that has since then been fixed.

  1. There is no explanation for Table 4 and Figure 4 in the text. Please add an explanation where appropriate.

That has been fixed, please let me know if the new caption and image (the other reviewers had issues reading figure 4) are sufficient.

Reviewer 2 Report

1.              The quality of Figures 3 & 4 needs to be improved. It is currently not readable

2.              Please provide stats on model accuracy and model validation efforts

3.              Please explain the significance and novelty of your research more clearly. What are the contributions of this research? 

4.              Please explain actionable and significant outcomes from this research

5.              Please explain how you envision the result of this study will be used and applied in practice? What is the impact and how would you quantify that?

Author Response

Reviewer #2:

  1. The quality of Figures 3 & 4 needs to be improved. It is currently not readable

Sincerest apologies, please let me know if the adjusted figures are still not readable.

  1. Please provide stats on model accuracy and model validation efforts

LSTM CCA as per figure 3 has an F1 score of 1.0 for tracking all five targets the dataset provides. I’ve published a few conference papers and a journal paper on many different approaches that have been taken with the same dataset, from CNN based to more traditional models, but this approach is the only to have achieved 100% accuracy with the dataset.

The samples used are only roughly 6 minutes of data, but are used for the same objectives as the other papers. The primary focus on this research is the explainability, and as such the model used is the one that has performed the best to gain the best insights into how the P-RF data impacts decision making. I can link a few papers if you wish but they are referenced in this paper.

  1.  Please explain the significance and novelty of your research more clearly. What are the contributions of this research? 

Passive-RF research that isn’t focused on radar is essentially fringe research, owing to a lack of other approaches in utilizing what is basically background “white noise”. While my group has done some research with its usage for detection and tracking of both human and vehicle targets, we have never had any quantifiable understanding as to how the I/Q data is utilized. This research presents insights into a blackbox approach that can accurately utilize the I/Q data to identify and track vehicle targets. In doing so, we are able to gain an a level of understanding as to how P-RF data impacts the decision making process of the blackbox approach.

  1. Please explain actionable and significant outcomes from this research

Obviously, it is difficult to quantify more subjective measures such as the visualization of neuron activation from a blackbox model. However, with what few papers that have been published on P-RF based detection, none of them have provided explainability beyond describing the general blackbox that is used. I personally have my doubts that the explainability aspect will have anything directly relevant to current research for similar objectives, owing to the choice of modality being rarely used, but at the very least it shows a more quantifiable impact that using P-RF data for this application has. On the more actionable end, it provides a cheaper (passive RF needs less equipment) alternative towards how detection and tracking can be pursued on a more urbanized scale.

For me personally, it provided confirmation beyond the “result” of adding the P-RF data to solve the same problem, and in a more intuitive manner. Being able to correlate how and when P-RF data is used with respect to the scenario provides greater insights and suggestions for wider scale data collection for similar applications. In the summer, I’ll be doing work for AFRL which will include scenario design, and this kind of information helps me decide how and what I should test and design for.

  1. Please explain how you envision the result of this study will be used and applied in practice? What is the impact and how would you quantify that?

Candidly speaking, the usage of P-RF data (passive I/q data that isn’t being transmitted but just recorded) that isn’t passive radar is honestly pretty fringe from what literature I’ve been able to find on the subject that isn’t mine, so I don’t have high hopes the approach will be utilized anytime soon. Its been fringe enough that my research group is actually actively pursuing a patent right now for our approach at detection with I/Q data. So, I’m not entirely certain if the research will be economically or technologically viable for wider scale use at this time.

Should a much larger project that aims to utilize P-RF data in the same way come up in the future, with respect to collecting real data on a much wider scale, the best I can honestly hope for in the immediate future is probably just a minor footnote in how such data has been utilized for comparison research.

For applications, given P-RF is considerably less “invasive” compared to other methods such as an Amazon Echo listening in and giving your data to the CIA or whatever, I would like to think that this might be utilized for smart home or possibly security applications. My research group has been focused on human detection as a whole, while mine is more focused on vehicle detection, so likely surveillance if my research is utilized.   

Reviewer 3 Report

This paper is based on two (LSTM-CCA) models to achieve sensor fusion, including passive radio frequency (P-RF) data and electro-optical (EO) data. It provides a higher performance explainable model to complete the task of differentiating to different targets. However, there are not enough algorithm results to prove the correctness of the authors' explainable model, and how the different data models will impact performance. Inhere, I have several concerns that I believe should be addressed.

Comments:

1. The description of the scene in Figure 1 requires more details. For example, does the triangle correspond to the car? According to the description of the ESCAPE dataset, it contains a variety of car types, which requires corresponding details on the picture or in the description. In addition, the trapezoidal and semicircular symbols also need more details. Please show the figure and corresponding descriptions as carefully as possible to help readers quickly understand the scene.

2. The link between input and output memory in Figure 2 is a bit of a logical problem. Why the View #1 and View #2 continue to bridge using memory when they can be linked directly to the CCA layer? This is a conflicting design, and what does the mean for a connection between View #1 and View #2? Finally, Output memory and the CCA layer form a loop, but what does it mean? I would suggest the author update Figure 2 which makes it more clear logical. Meanwhile, the description of Figure 2 is missing from the paper, it is suggested that the author can match the content of the entire Section 3.3 with Figure 2.

3. For the presentation of the classification results, please add the ROC curve and PR curve, which will make the results easier to understand. Meanwhile, Figure 3 is too blurry to see the specific values. Please authors update Figure 3, and suggest adding the ROC curve of the algorithm and PR curve to show the performance of the algorithm, rather than only the Normalized Confusion Matrix (Moreover, the authors missed explaining "CM" as a Confusion Matrix, in the figure 3 title.)

4. The description of Figure 3 is too simple, and it is difficult to understand why the algorithm has obtained a perfect score. Does it mean that the classification effect of the confusion matrix is 100% accuracy?

5. Figure 4 has insufficient resolution and the description is missing. I would authors can carefully check the paper that makes sure all figures have been described.

6. In Section 4.2, does the author want to discuss the attention mechanisms of the four types of data (the DOF-EO, EO-CCA variates, P-RF histograms, and the RF-CCA variates) here? The author's explainable AI model is not a good expression of how data fusion can achieve better performance. The results of the weight experiment are not proved by the subsequent comparison results.

7. In the experimental section, the contribution of each data module in this algorithm is not evaluated. Please use an ablation experiment to verify the impact of each data module on the accuracy. It is recommended that the author update the results of scenarios 1 and 2C. Not only to show the weight result, but also to add the specific performance of the algorithm for each data model. This way proves the correctness of the explainable weight results.

8. In Section 5, there is a lack of comparison results of related works. The authors use an open-source dataset, so it is necessary to show a comparison table with results, and then analyze the differences between the method and other work. Meanwhile, this section should discuss the advantages and disadvantages of different methods. Finally, the author should prove the novelty of the method.

Round 2

Reviewer 2 Report

1.              The quality of Figures 3 & 4 are slightly improved, but still needs to be further improved. In general, most of the Figures have poor quality

2.              In Figure 3, you are providing confusion matrix with perfect scores. Does that sound realistic and practical? how do you explain this perfect model. Providing more details of your model validation and accuracy is needed

3.              Please explain the significance and novelty of your research more clearly. What are the contributions of this research? 

4.              Please explain actionable and significant outcomes from this research

5.              Please explain how you envision the result of this study will be used and applied in practice? What is the impact and how would you quantify that?

Author Response

  1. The quality of Figures 3 & 4 are slightly improved, but still needs to be further improved. In general, most of the Figures have poor quality

My sincerest apologies, I have tried several tricks for attempting to improve the quality, but within the confines of LaTeX this is best to improve the quality of the figures. I can’t really increase the size of the figures without them going off the page or ending up in strange places. Which I’m assuming is a higher priority, but if its not a problem for the end publication, I’ll be more than happy to leave them at full size, which is at this point the only way to improve the quality.

  1. In Figure 3, you are providing confusion matrix with perfect scores. Does that sound realistic and practical? how do you explain this perfect model. Providing more details of your model validation and accuracy is needed

I am starting to think that figure 3 is seriously detracting from the point I’m trying to make in that section, so I’ve removed the figure from the paper.

Obviously, no one in their right mind thinks 100% is either practical or realistic. Within the confines of the ESCAPE dataset and the scenarios tested, this is the highest performance I’ve gotten (runner ups being in the 0.95 and 0.9 range), as the CCA layer to the model appears to have pushed the F1 score by the last few percentages needed for a perfect score. The focus of the paper is on the explainability results, and for obvious reasons having the highest performing model instead of the lower performance models is ideal for explanations. Evaluating the performances over three different scenarios is to prevent overfitting and to ensure that the model’s approach is more robust, but the combined three scenarios at the end of the day is a little less than 6 minutes of samples.

As for model validation and accuracy, I used the standard cross validation, with 30% of the data used for validation. As the paper was originally longer, and is still at 17 pages, I was trying to remove unnecessary details for the sake of keeping the paper on point and concise.

  1. Please explain the significance and novelty of your research more clearly. What are the contributions of this research? 

This research provides insights into how I/Q data is used within a fusion model for vehicle tracking and detection.

I/Q data is normally used for active broadcasts of data, but instead of the vehicles or anything else broadcasting signals, the I/Q data collected is just what’s in the background, essentially white noise. Most tracking and detection applications rely on active RF or radar based approaches. Moreover, as our research has been the only research into using this for tracking, we’ve only been able to use Blackbox based approaches. The explanations provide transparency and insights as to how this data, which is otherwise unintuitive and near impossible to use by traditional methods is the main contribution of the paper.

  1. Please explain actionable and significant outcomes from this research

My answer hasn’t really changed since the last round of reviews.

  1. Please explain how you envision the result of this study will be used and applied in practice? What is the impact and how would you quantify that?

My answer hasn’t really changed since the last round of reviews.

Reviewer 3 Report

The paper has been improved after the first round of reviews. However, many issues still need to be processed. The main problem of the paper is an uncertain novelty, the authors explained no one had implemented the same modality for fusion and published insights into how the P-RF data is used.

I am just confused about this answer, such as in the authors' response to comment 6, I asked how the fusion can achieve better performance. The authors don't have a straight answer, just said:  It is the insights of the greedy algorithm that are the bigger focus of the actual paper.

I can understand authors want quickly publish the paper, but it doesn't mean quick response comments can replace the work for improving the paper. I didn't see any effort work from authors in the second round of paper submission.
